# Experimental Study of Black Cotton Soil Stabilization with Natural Lime and Pozzolans in Pavement Subgrade Construction

Zihong Yin [1], Raymond Leiren Lekalpure [1,*] and Kevin Maraka Ndiema [1,2]

1  School of Civil Engineering, Southwest Jiaotong University, Chengdu 610031, China; 71yzh@163.com (Z.Y.); marakakevin@yahoo.com (K.M.N.)
2  Department of Civil Engineering, School of Engineering, University of Eldoret, Eldoret 30100, Kenya
*  Correspondence: raymondleiren@gmail.com; Tel.: +254-7287-38755

**Abstract:** This study explores the engineering characteristics of Black cotton soil (BCS) stabilized with natural lime, volcanic ash (VA), and their mixtures. Based on the available literature, the stabilization of VA-BCS is limited. Laboratory tests conducted on stabilized BCS include the Atterbeg limits, the proctor test, the swell percent test, and the California bearing ratio (CBR). The results showed that adding VA and lime greatly improves the engineering characteristics of BCS. BCS stabilized with a mixture of VA and lime showed superior results. Adding 3% lime with 20% VA increased natural CBR values 10.76 times, reduced plasticity by 29%, and reduced swell percent by 88%. Stabilized BCS with 3% lime + 20% VA meets the minimum swell, plasticity, and strength requirements; thus, it can be used as an alternative to cutting and filling.

**Keywords:** black cotton soil; volcanic ash; road construction materials; subgrade; free swell; plasticity index; California bearing ratio (CBR)





## 1. Introduction

The construction of new railways, highways and rural roads has rapidly developed in Kenya [1,2]. This development has led to heavy construction of infrastructure projects, which has increasingly faced engineering difficulties due to black cotton soil [3]. Expansive soils form a significant percentage of soils in Kenya [4]. The depth of BCS in Kenya ranges from 0.2 to 0.3 m [5]. BCS is characterized by high swell potential, low shear, low bearing capacity, and excessive compression and dispersion [6]. Clay minerals including chlorite, vermiculite, illite and montmorillonites were found to be abundant in BCS [7]. The high content of clays minerals is the cause of high expansion in BCS [8]. BCS forms deep cracks in dry seasons and expands in wet seasons; these changes, shrinking and swelling, cause road deformation [9]. Therefore, untreated BCS does not meet the material requirements for the placement of subgrade soil and must be cut and filled with suitable material. Various methods of dealing with BCS in Kenya include excavation and replacement with suitable material, stabilization with Portland cement, confining BCS under improved subgrade by using a 300 mm capping layer or entire re-alignment of the proposed road corridor to avoid BCS areas [5]. However, substantial use of these methods dramatically increases construction budgets. Thus, it is vital to find a readily available local replacement material to deal with BCS.

Mixing the materials with inherent binder properties with unstable soils to increase the engineering characteristics of such soils is referred to as soil stabilization. Stabilization greatly reduces the plasticity and swelling potential of BCS, which improves its strength and durability [10–12]. Admixtures with, e.g., Portland cement, rice husk ash, hydrated lime and cement kiln dust have been utilized as stabilizers [13–16]. Other methods include the use of bagasse ash [17]. Al-Mukhtar et al. [18,19] studied lime–soil reactions for both

short and long term durations, and Al-Rawas [19] researched the effects of lime and artificial pozzolana on black cotton soil by use of Atterberg limits, swell potential, swell pressure, CBR and unconfined compression tests (UCS). They found that lime and artificial pozzolana greatly improve the engineering characteristics of BCS. F Sitepu et al. [20] investigated the engineering characteristics of VA stabilized BCS by means of the Atterberg limit, CBR and UCS and found that VA increased UCS and CBR, and decreased the liquid limit (LL) and plasticity index (PI), but not significantly. A combination of lime and VA presented a better improvement. Karatai et al. [3] studied the influence of rice husk ash in the treatment of BCS and found that the addition of 20% rice husk ash with 2% lime increases BCS CBR by 800%, decreases soil plasticity by 90% and decreases free swell by 70%. However, rice husk ash is limited only to rice-growing areas.

Volcanic ash (VA), which forms due to volcanic activity, is abundantly found in Kenya [21]. It is rich in minerals, volcanic fragments and rocks and can be used to stabilize BCS. The pozzolanic properties of VA enables it to bind with BCS clay particles and reduce clay swelling. Thus, VA-stabilized BCS decreases the expansive potential and the PI, and increases durability and strength [22]. The utilization of VA as a BCS stabilization agent is limited in Kenya. Zhang et al. [7] studied a combination of VA and BCS for use in the production of hollow clay bricks and found that the bricks exhibit an excellent compressive strength of more than 60 Mpa. This paper was written to establish the viability of the use of VA found in Kenya to stabilize BCS. The effects of natural pozzolana and natural lime on the plasticity, swell and strength of stabilized BCS were investigated.

## 2. Materials

The materials used in these experiments included Black Cotton Soils, Volcanic Ash soils and Natural lime.

### 2.1. Black Cotton Soils (BCS)

The BCS used in this study was obtained at Km 20 + 080 along Kapsokwony-Kopsiro-Namwela road, 20 Km from Kimilili town in Western Kenya. Figure 1 presents a sample of BCS used in this study. Chemical and engineering properties, in terms of Atterberg limits based on BS 1377: Part 2:1990 [23], are presented in Tables 1 and 2, respectively. With a CBR value of 1.7, a liquid and plastic limit of 70.7 and 35.7, respectively, and a plasticity index of 35, the obtained soil is classified as black cotton soil according to the Ministry of Public Works Kenya: Road material design manual [5]. Figure 2 illustrates the particle size distribution curve of the obtained BCS.

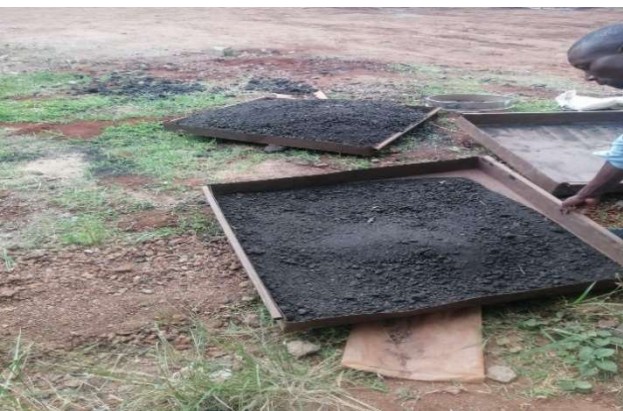

**Figure 1.** BCS material sample.

**Table 1.** Engineering properties of BCS.

| Properties | Test Values |
|---|---|
| LL (%) | 70.7 |
| PL (%) | 35.7 |
| PI (%) | 35 |
| MDD (g/m$^3$) | 1449 |
| OMC (%) | 34 |
| CBR (%) | 1.7 |
| Linear Shrinkage Limit (%) | 16.4 |
| Free Swell Index (%) | 3 |
| Plastic Modulus (%) | 1680 |
| Grading% passing sieve size 0.425 mm | 48 |

**Table 2.** Volcanic ash and black cotton soil chemical properties.

| Properties | Volcanic Ash (VA) | Black Cotton Soil (BCS) |
|---|---|---|
| CaO | 10.26 | 1.68 |
| $Fe_2O_3$ | 12.24 | 9.22 |
| MgO | 11.42 | 0.97 |
| $K_2O$ | 1.29 | 1.03 |
| $SO_3$ | 0.08 | - |
| MnO | 0.17 | 0.26 |
| $AL_2O_3$ | 13.46 | 17.04 |
| $Na_2O$ | 2.66 | 0.69 |
| $P_2O_2$ | 0.49 | 0.02 |
| $TiO_2$ | 2.63 | 0.91 |
| $SiO_2$ | 43.54 | 50.28 |
| Loss on ignition | 0.46 | 18.02 |

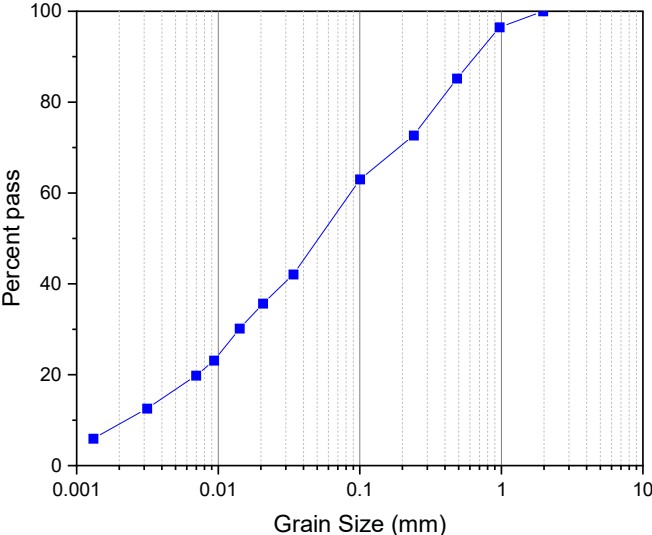

**Figure 2.** Natural BCS particle size distribution curve.

From the particle size distribution curve:

$$C_\mu = \frac{D_{60}}{D_{10}} = 32.79 \tag{1}$$

$$C_C = \frac{D_{30}^2}{D_{60}D_{10}} = 0.65 \tag{2}$$

$$D_{60} = 0.1004 \tag{3}$$

$$D_{30} = 0.0142 \tag{4}$$

$$D_{10} = 0.0031 \tag{5}$$

With coefficient of uniformity > 15 and coefficient of curvature < 0.5. The sampled BCS used in this study can be classified as Gap graded according to the BS EN ISO 14688-2:2018 classification, based on particle size distribution [24].

### 2.2. Volcanic Ash

The natural volcanic ash (VA) sample used in this study was obtained at the floor of the Great Rift Valley near Ewaso Kendong town situated between Mt. Suswa and Mt. Longonot, 65 km Northwest of Nairobi. The sample was scooped 0.2 m below the topsoil. Figure 3 shows VA sample used in this study. The VA's chemical properties are shown in Table 2.

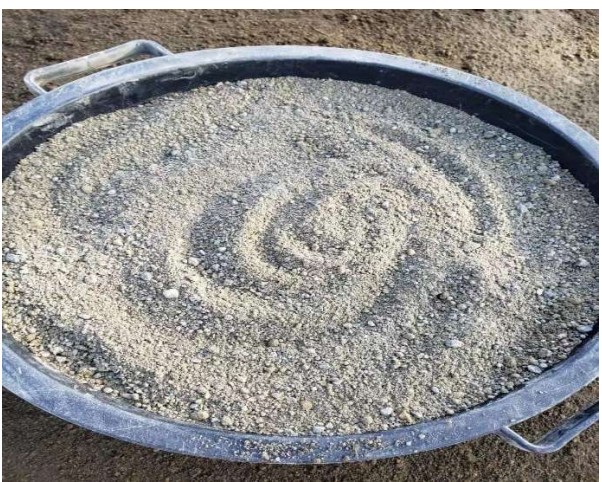

**Figure 3.** Volcanic ash material sample.

### 2.3. Hydrated Lime

Hydrated lime was procured from a local store near the laboratory.

## 3. Methods

### 3.1. Combination Scheme and Sample Preparation

Volcanic ash (VA) and black cotton soil (BCS) were air-dried and grounded to pass through 0.75 and 0.5 mm sieving openings. Both VA and BCS were oven-dried for 24 h at 105 °C temperature. The samples were then weighed and thoroughly mixed with different lime, VA and BCS combination dosages, after which they were ready for testing. The design scheme for BCS stabilization is presented in Table 3.

### 3.2. Engineering Properties Measurements

The engineering characteristics of black cotton soil stabilized with volcanic ash and lime were assessed in this study, including a determination of the Atterberg limits, in addition to the swell potential proctor test and CBR.

### 3.3. Assessment of Plasticity

The plastic limit (PL) was measured per British Standard 1377-Part 2:1990 [23]. Samples of BCS and BCS with various mixtures passing through a 0.425 mm sieve were air-dried, mixed thoroughly with distilled water, and soaked for 24 h. PL is the moisture content at which a molded soil sample loop breaks when its diameter nears around 3 mm.

**Table 3.** The design scheme for BCS stabilization.

| No. | Volcanic Ash (%) | Hydrated Lime (%) | Mix Ratio VA:L | Total Stabilizer Content (%) | Notes |
|---|---|---|---|---|---|
| 1 | - | - | 0:0 | - | Neat BCS |
| 2 | - | 1 | 0:1 | 1 | - |
| 3 | - | 2 | 0:2 | 2 | - |
| 4 | - | 3 | 0:3 | 3 | Lime and BCS |
| 5 | - | 4 | 0:4 | 4 | - |
| 6 | 10 | - | 10:0 | 10 | - |
| 7 | 15 | - | 15:0 | 15 | VA with BCS |
| 8 | 20 | - | 20:0 | 20 | - |
| 9 | 25 | - | 25:0 | 25 | - |
| 10 | 15 | 1 | 15:1 | 16 | - |
| 11 | 20 | 1 | 20:1 | 21 | VA, lime and BCS |
| 12 | 15 | 3 | 5:1 | 18 | - |
| 13 | 20 | 3 | 20:3 | 23 | - |
| 14 | 15 | 4 | 15:4 | 19 | - |
| 15 | 20 | 4 | 5:1 | 24 | - |

The liquid limit (LL) is defined as the water content that changes soil from plastic state to liquid state. The Casagrande apparatus was used to determine LL as per the British Standard 1377-Part 2:1990 [23]. Samples were prepared by passing through a 0.425 mm sieve, oven-dried, mixed thoroughly with distilled water and soaked for 24 h. The soil water content at which 25 blows caused 13 mm closure of the groove at the top of the cub was used as the soil LL.

The plasticity index is the difference of the liquid limit and the plastic limit:

$$Plasticity\ Index\ (PI) = LL - PL \tag{6}$$

### 3.4. Assessment of Soil Compaction Characteristics

A proctor test was conducted to determine the maximum dry density (MDD) and optimum moisture content (OMC) as per AASHTO T180:2011 [25]. Black cotton soil (BCS) and BCS specimens with various additives were thoroughly mixed and stored for 24 h. A compaction test was performed for all 15 specimens to determine the MDD and OMC.

### 3.5. Assessment of Soil Bearing Strength

The California bearing ratio (CBR) was measured per the American Standard ASSHTO: T193:2003 [26]. Specimens were prepared at the OMC and MDD and stored for 24 h. Samples were compacted into three layers of CBR molds. A 2.5 kg rammer was used to apply 30 blows in every layer. After soaking for 7 days, a penetration test was conducted with a 1935 mm² area plunger at a 1.27 mm/min penetration rate. Values corresponding to 2.5 mm penetration were computed and recorded as the sample's CBR values.

### 3.6. Assessment of Swelling Soil Characteristics

Swell percent was determined using the same CBR apparatus as per ASSHTO: T193:2003 [26]. Figure 4 shows the soaked specimen fitted with a CBR swell tripod. The initial gauge reading was recorded as (R1) mm; the final gauge reading was recorded as (R2) mm after 7 days of soaking. The swell percent was calculated as follows:

$$Swell = \frac{(R1 - R2)}{116.4} \times 100\% \tag{7}$$

where 116.4 mm is the height of the specimen.

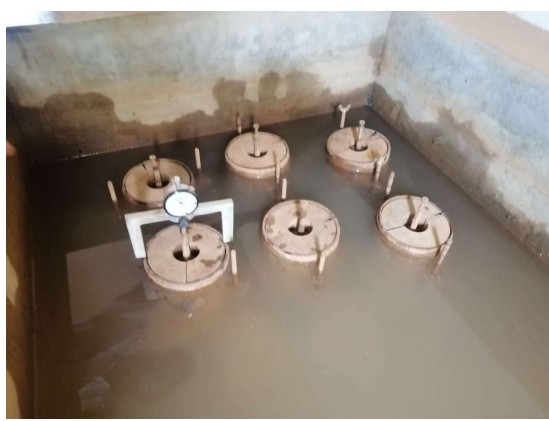

**Figure 4.** CBR swell tripod for measuring swell.

## 4. Results and Discussion

### 4.1. Effects of Volcanic Ash and Hydrated Lime on Plasticity

Table 4 presents the LL and PL of natural and stabilized Black cotton soil. The LL of BCS with the addition of 25% Volcanic Ash decreased by 6.9% from 70.7% to 65.8%. LL of BCS mixed with lime also showed a steady decrease in LL; the use of 4% lime-stabilized BCS reduced the LL from 70.7% to 64.3%. BCS samples mixed with 4% lime + 15% VA presented the largest decrease in the LL value from 70.7% to 53.9%, a 38% reduction.

**Table 4.** Liquid and plastic limits of neat BCS and with stabilizer.

| Sample No. | Samples | Liquid Limit (%) | Plastic Limit (%) |
|---|---|---|---|
| 1 | Neat BCS | 70.7 | 35.7 |
| | Lime (%) | | |
| 2 | 1 | 71.3 | 37.9 |
| 3 | 2 | 69.2 | 42 |
| 4 | 3 | 65.8 | 42.8 |
| 5 | 4 | 64.3 | 43 |
| | Volcanic ash (%) | | |
| 6 | 10 | 75.4 | 42 |
| 7 | 15 | 65.6 | 35.6 |
| 8 | 20 | 65 | 35 |
| 9 | 25 | 65.8 | 40 |
| | Mix stabilizer | | |
| 10 | 15% VA + 1% L | 64.4 | 32.9 |
| 11 | 20% VA + 1% L | 69.5 | 39.7 |
| 12 | 15% VA + 3% L | 58.2 | 32 |
| 13 | 20% VA + 3% L | 56.9 | 32 |
| 14 | 15% VA + 4% L | 53.9 | 33.9 |
| 15 | 20% VA + 4% L | 56.9 | 35.6 |

According to Figure 5a,b, for all the studied materials, a steady decrease in the plasticity index was observed. This is attributed to the pozzolana properties of both lime and VA. Pozzolans react with expansive soils to form compounds with cementitious characteristics; this leads to steady a reduction in the plasticity of BCS.

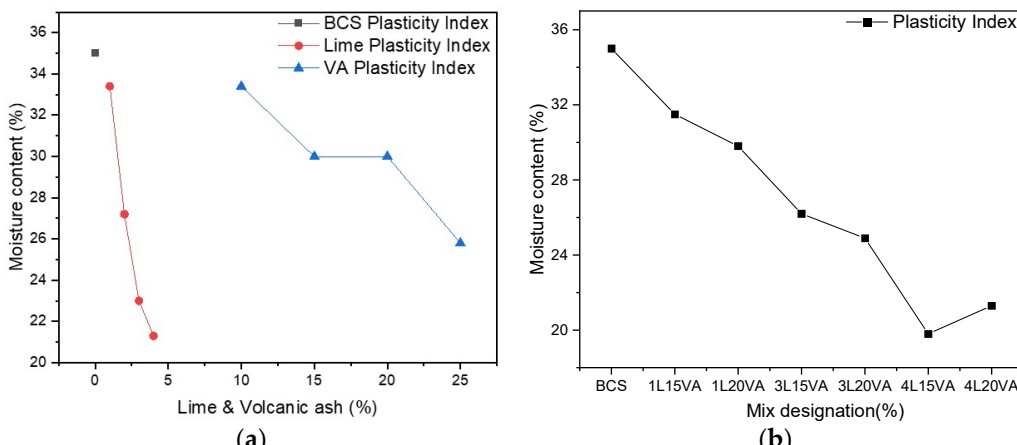

**Figure 5.** Effects of (**a**) lime and VA and (**b**) Lime–VA combined stabilizer on the plasticity of BCS.

Figure 5b presents the effect of lime, VA and their mixtures on the plasticity index of natural BCS. BCS mixed with 4% lime, 25% VA, and a mixture of 4% lime + 20% VA decreased the plasticity index from 35% to 21%, 26% and 21%, respectively. According to the Ministry of Public Works Kenya: Road material design manual [5], the plasticity index of subgrade layer material should not exceed 50%. The sampled BCS used in this study satisfies the code plasticity index requirement but does not meet the swell and CBR requirements. BCS stabilized with 3% lime + 20% VA reduced the PI by 1.2 times, to 24.9%.

### 4.2. Effects of Stabilizers on Compaction Characteristics

Figure 6 presents the influence of lime and volcanic ash (VA) on the optimum moisture content (OMC) and maximum dry density (MDD) of black cotton soil (BCS). As observed in Figure 6a, the addition of lime caused a gradual decrease in the MDD and a nearly linear increase in the OMC. Similar behavior was observed by other researchers [3,21]. The immediate decrease in MDD due to the addition of lime can result from particles' aggregation caused by lime. These particles occupy larger spaces and affect soil grading; further drops in density with the addition of lime can be attributed to the replacement of soil particles with lime particles of comparatively low specific value.

In Figure 6b, the MDD increased while the OMC decreased with the addition of volcanic Ash (VA) from 0% to 25%. Hossain et al. [22] similarly observed the same behavior. According to their explanation, an increase in MDD with the gradual addition of VA is attributed to VA's lower affinity for water. Increased MDD is an indication of improved soil strength characteristics.

### 4.3. Effects of Volcanic Ash and Hydrated Lime on the California Bearing Ratio and Expansion Ratio

The CBR and expansion ratio of treated black cotton soil after 7 days of soaking are presented in Figure 7. The lime-stabilized BCS CBR values increased with the increasing of the lime content from 1% to 4%. The CBR values increased from 1.7% to 19%. The expansion ratio of lime-stabilized BCS decreased from 4% to 0.4%, with 4% lime. This behavior is attributed to cation and pozzolanic reactions between BCS and lime.

The CBR values of the Volcanic Ash (VA)-stabilized soils increased from 1.7% to 7% with 25% VA. According to the Ministry of Public Works Kenya: Road material design manual, a minimum of 8% CBR is required for the class 2 (S2) subgrade. Subgrade class S2 has a CBR range of (5–10) with a medium of 7.5. The code recommends that subgrade soils with this range of CBR be further treated with cement. Table 5 presents the subgrade soil classification in Kenya. Thus, BCS stabilized with VA alone is not sufficient to lay a subgrade. The expansion ratio decreased 2.7-fold from 4% to 1.5% with the Addition of 25% VA.

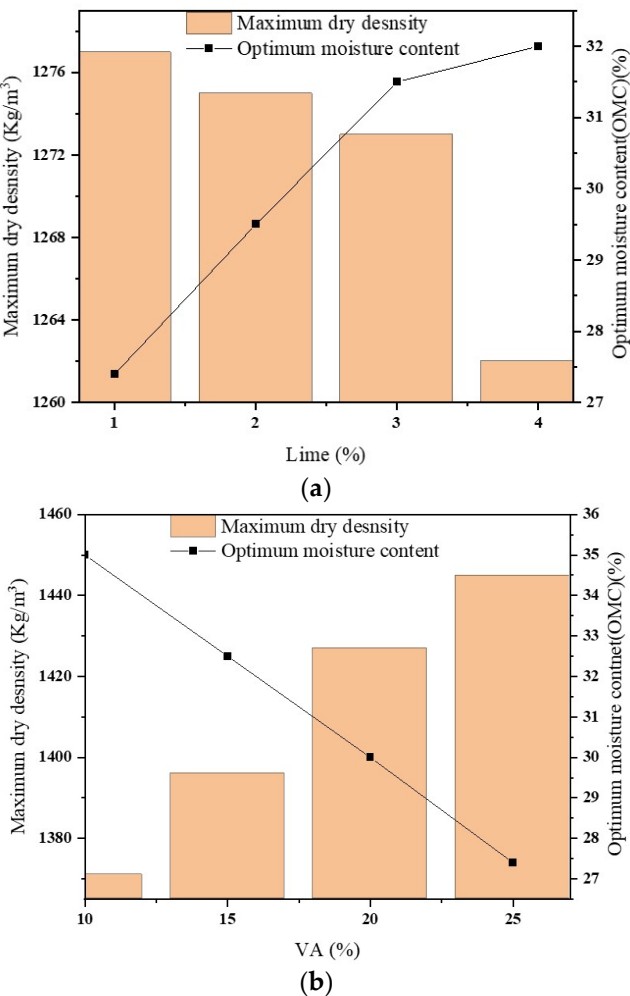

**Figure 6.** Effects of (**a**) lime and (**b**) VA on OMC and MDD of black cotton soil.

**Table 5.** Subgrade soil classification in Kenya [5].

| Subgrade Class | CBR Range (%) | Median |
|:---:|:---:|:---:|
| S1 | 2–5 | 3.5 |
| S2 | 5–10 | 7.5 |
| S3 | 7–13 | 10 |
| S4 | 10–18 | 14 |
| S5 | 15–30 | 22.5 |
| S6 | >30 | - |

The lime and VA mix stabilizer showed a significant increase in CBR values. The CBR values increased from 1.7% to 25% with the addition of composite stabilizer 4% lime + 20% VA. Expansion was reduced from 4% to 0.3% with the addition of 4% lime + 20% VA.

Table 5 shows the CBR requirement of subgrade material in Kenya. According to Figure 7c, BCS stabilized with 3% lime + 20% VA with a 20% CBR ratio satisfies the requirement and can be used as a subgrade material for subgrade class S2 (CBR range (5–10)) to S5 (CBR range (15–30)).

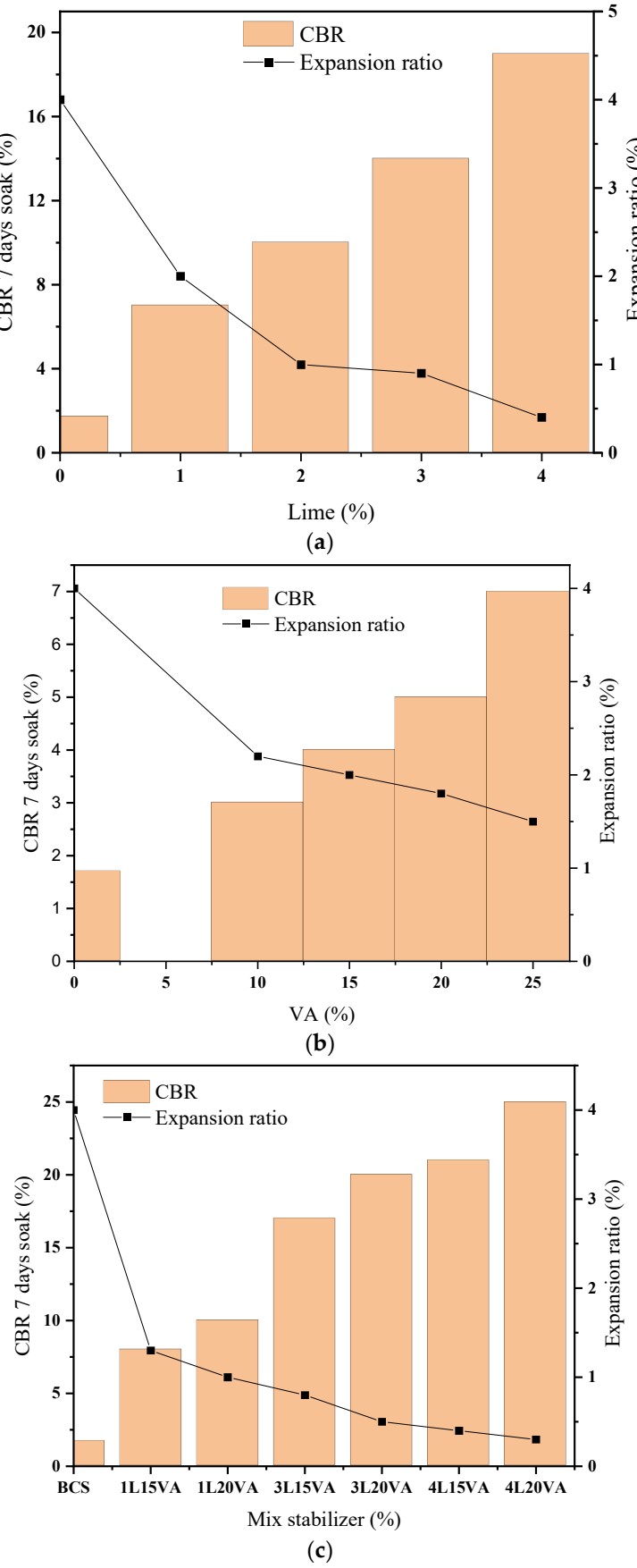

**Figure 7.** Effects of (**a**) lime, (**b**) volcanic ash and (**c**) a combination of volcanic ash on CBR and the swell percent value of BCS.

## 5. Conclusions

Atterberg limits, compaction, CBR and swell percent tests were used in this study to investigate the engineering characteristics of black cotton soils stabilized with volcanic ash and lime as stabilizing agents. The following conclusions were drawn:

- BCS stabilized with 3% lime + 20% VA exhibited enhanced mechanical properties such as an increase in the CBR and MDD, which indicated an increase in strength.
- The addition of 3% lime + 20% VA increased the BCS CBR values significantly, by 10.76 folds, reduced the swell potential by 88%, and reduced the plasticity index by 29%, thus meeting the standard requirement for Class S2 and above pavement subgrade materials according to Kenya's national road design manual.
- This study found that both lime and VA significantly reduced the plasticity of BCS.
- This study found that volcanic ash sourced in Kenya mixed with lime in a 3% lime + 20% VA combination can be used as an effective BCS stabilizer.

In line with the above conclusions, the authors recommend the use of lime and volcanic ash in a 3% lime + 20% VA combination to address the problem of black cotton soil in areas where volcanic ash is available as an alternative to cutting and filling or an over reliance on Portland cement, in order to reduce road infrastructure construction budgets.

**Author Contributions:** Conceptualization, R.L.L.; Supervision, Z.Y.; Writing—original draft, R.L.L. and K.M.N.; Writing—review & editing, Z.Y. and K.M.N. All authors have read and agreed to the published version of the manuscript.

**Funding:** This study was supported by the National Key research and Development Project of China (Grant No. 2016YFC0802203).

**Institutional Review Board Statement:** Not applicable.

**Informed Consent Statement:** Not applicable.

**Data Availability Statement:** The authors confirm that the data supporting the findings of this study are available within the article.

**Acknowledgments:** The authors thank Michael Rono, Moses Ogire, Kapsokwony-Namwela & Malinda Road Project for allowing us to conduct some experiments at the Project Laboratory.

**Conflicts of Interest:** The authors declare that there is no conflict of interest regarding this submission.

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
