# Peer review of "Experimental Study of Black Cotton Soil Stabilization with Natural Lime and Pozzolans in Pavement Subgrade Construction"

_coatings, doi:10.3390/coatings12010103_

Round 1

Reviewer 1 Report

Manuscript

Experimental Study of Black Cotton Soil Stabilization with Natural Lime and Natural Pozzolans as an Alternative to Cut and Fill in Pavement Subgrade Construction

The article is good in its structure and must be highly valued by the construction engineering community.

But it seems to be not within the scope of this journal, authors may submit other related journals,

Please do not think this is against the technical quality of your work. However, the scope and impact of the paper is not of general interest in the relevant reading community.

Hence, reject reviewing for Coatings Journal.

With Regards,

Author Response

Thank you for your comment.

Reviewer 2 Report

This study explores engineering characteristics of Black cotton soil (BCS) stabilized with natural lime, volcanic ash(VA), and their mixtures. Based on the available literature, VA-BCS stabilization is limited. The paper is very well organised. English writing requires a general review. The bibliography should be improved. Also, it is important to systematically avoid mass citations, for example [10-12], [13-16]. 

Literature can be explored better. Include a more recent and relevant bibliography. Some suggestions: 
https://doi.org/10.3390/en13184970
https://doi.org/10.3390/en12152922
https://doi.org/10.3390/en14165200
https://doi.org/10.3390/en13184970
1016/j.enbuild.2014.07.039
1016/j.energy.2021.120482

Adjust the style and size of the part as it is now versatile. This needs to be changed.
Expand the discussion to an article.
It is worth proposing 2-3 conclusions that can be used in practice.

Reviewer 3 Report

Title:  Experimental Study of Black Cotton Soil Stabilization with Natural Lime and Natural Pozzolans as an Alternative to Cut and Fill in Pavement Subgrade Construction

Journal: Coatings

I evaluate the lectured article as a scientific contribution with significant potential for implementation in technical practice with a synergistic environmental and economic dimension. However, in order to meet my expectations of a quality scientific article, the following requirements and recommendations need to be incorporated.

Mandatory requirements and recommendations

LNSA (Line Number of the Scientific Article) 2-4…. I recommend modifying the title of the article, especially removing it from the title cut and fill (it appears in the text of the article in addition to the title only in lines 20 and 36). You should also consider removing the repetitive word Natural. As an illustrative example, I would like to mention e.g. next name: Experimental Study of Black Cotton Soil Stabilization with Natural Lime and Pozzolans in Pavement Subgrade Construction.

LNSA 13… volcanic ash(VA)...Insert a missing space, this requirement applies to the whole article under consideration.

LNSA 17… Adding 3%lime with 20%VA increased natural CBR values by 1076%... Missing spaces should be inserted (replace 3%lime with 20%VA with 3% lime with 20% VA) and replace the 1076% percentage increase with a multiple value.

LNSA 27-28... Kenya.[1,2] This development has led to heavy construction of infrastructure projects, which has increasingly faced engineering difficulties due to black cotton soil[3].… Unify the writing of references to information sources at the end of sentences.

LNSA 52, 56… F Sitepu et al[21]investigated,… and Karatai et al. [3]… Throughout the document, it is necessary to unify the form of writing the abbreviation of the Latin term Et alii (et al or et al.).

LNSA 90…MDD(kg/m3)… replace MDD(kg/m3).

LNSA 91, 94… Volcanic Ash(VA) Black Cotton, Soil(BCS), Volcanic ash (VA) and black cotton soil (BCS)…Unify the writing of abbreviations in the Table 2. and text of the article.

LNSA 104, 110…British Standard 1377 Part 1 (1990)…Currently valid BS 1377-1:2016 Methods of test for soils for civil engineering purposes - General requirements and sample preparation. It is necessary to explain what led to this disproportion in the listing of BS 1377-1, explain whether the laboratory tests were performed at the time of BS 1377-1: 1990, or it is just an incorrect reference to the literature or. whether it was caused by other circumstances.

LNSA 121-122… California bearing ratio (CBR) was measured per the American Standard ASSHTO: T180…The following standards should currently apply AASHTO T 193:2013 Standard Method of Test for The California Bearing Ratio and . AASHTO T 180:2021 Standard Method of Test for Moisture–Density Relations of Soils Using a 4.54-kg (10-lb) Rammer and a 457-mm (18-in.) Drop.  It is necessary to state from which years the American Association of State Highway and Transportation Officials (AASHTO) standards were used and to consider them in references.

LNSA 124… After soaking for 7 days, penetration test with a 1935mm2 area plunger...Remove typos in the upper index, ie replace 1935mm2 with 1935mm2.

LNSA 145, 147... Figure 2. Effect of (a) lime, (b) VA… In the Figures 2. Effect of (a) lime, (b) VA I recommend to show the correlation dependences of the change in moisture content (%) on the amount added (%) of Lime resp. Volcanic Ash.

LNSA 147...Valcanic Ash (%)…remove the typo “valcanic” shown in Fig.2 b).

LNSA 167, 169… Figure 3. Effects of (a) lime and VA on OMC and MDD of black cotton soil…The names of the y-axis on the right in Figures 3 (a), 3 (b) should be written in accordance with the names of the other axes, i. with a capital letter at the beginning of the axis name.

LNSA 171…Figure 3. Effects of (a) lime and VA on OMC and MDD of black cotton soil  to replace  Figure 3. Effects of (a) lime and (b) VA on OMC and MDD of black cotton soil.

LNSA 180-182… Road material design manual a minimum of 8% CBR is required for the class 2(S2) subgrade. ThusThus BCS stabilized with VA alone is not sufficient to lay a subgrade...Briefly explain what the class 2 (S2) subgrade means and add the required value for the class 1 (S1) subgrade.

LNSA 203... Remove the subdivision Table 5. Subgrade soil classification in Kenya [5] on page 10 and 11.

LNSA 237… Z. W. S. S. Z. Yin...replace with the correct authors ZF Wang, SL Shen, ZY Yin, YS Xu.

LNSA 254, 256, 258…M. Al-mukhtar, A. A. Al-rawas, A. W. Hago, and H. Al-sarmi...to replace Al-Mukhtar, Al-Rawas, Al-Sarmi.

Optional recommendations for incorporation, if the authors have the relevant materials available

LNSA 28-30... Expansive soils form a significant percentage of soils in Kenya[4]. The depth of BCS in Kenya ranges from 0.2m to 0.3m[5]... Due to the fact that, as a European reviewer, I have not dealt with BCS in more detail so far, it would be appropriate to include photographs of the assessed soil in this section or. in the part dealing with the methodology of determining PS resp. CBR.

LNSA 71-72…Materials used in these experiments included Black Cotton Soils, Volcanic Ash soils and Natural lime… Similar to LNSA 28-30 would be appropriate to include photographs of the assessed soil in this section or. in the part dealing with the methodology of determining PS resp. CBR.

LNSA 76, 104, 110… BS 1377: Part 2:1990, British Standard 1377 Part 1 (1990)…BS 1377-2:1990 Methods of test for soils for civil engineering purposes - Classification tests.

LNSA 88-89… Figure 1. Natural BCS Particle size distribution curve... ISO 14688-1:2017 Geotechnical investigation and testing — Identification and classification of soil — Part 1: Identification and description. ISO 14688-1:2017 specifies the rules for the identification and description of soils and is intended to be read in conjunction with ISO 14688‑2 (ISO 14688-2:2017 Geotechnical investigation and testing — Identification and classification of soil — Part 2: Principles for a classification), which outlines the basis of classification of those material characteristics most commonly used for soils for engineering purposes. The relevant characteristics could vary and therefore, for particular projects or materials, more detailed subdivisions of the descriptive and classification terms could be appropriate.

Soil fraction is a group of soil particles with a diameter within a defined interval. Definitions of the fraction interval may vary depending on the classification standard used; USCS (Unified Soil Classification System – established by ASTM) and ISO 14688 classifications are usually applied in Central Europe.

If the authors are familiar with the ISO (International Organization for Standardization) standards, it would be appropriate to give the equivalent name of BCS according to ISO.

LNSA  128… Swell per cent was determined using the same CBR apparatus as per ASSHTO: 180…It would be appropriate to add quality photos to the article “the same CBR apparatus as per ASSHTO:180”.

LNSA 145-152… Figure 2. Effect of (a) lime, (b) VA and (c) lime-VA combine stabilizer on the plasticity of BCS…I recommend using the same scale range for the y-axis in Figures 2. a) to c) for the possibility of fast visual comparison.

LNSA 182-183…The expansion ratio decreased by 62.5% from 4% to 1.5% with the Addition of 25%VA... I recommend considering replacing the percentage decrease in the expansion ratio (62.5%) with a multiple expression (a 2.7-fold decrease resp. ... decrease 2.7 times).

LNSA 205-216...I consider the 10-line Conslusion to be insufficient for a quality scientific contribution in a renowned scientific journal. I recommend the authors to elaborate in more detail the potential of their objectified results into technical practice in Kenya and other countries with the occurrence of similar countries. It would also be appropriate to confront their research results with the works of renowned foreign authors. The mathematical expressions of correlation dependences of interest variables presented in Figure 2. Effect of (a) lime, (b) VA could also be presented in this section..

In view of the high potential of early implementation of objectified research results into technical practice in the conditions of Africa, I recommend that the authors quickly incorporate Mandatory requirements and recommendations. From the Optional Recommendations for Incorporation, if the authors have the relevant materials available, I recommend giving priority to the requirements presented in LNSA 71-72, LNSA 88-89, LNSA 128 and in particular LNSA 205-216. Based on the above facts, I will process the report of the revised version of the paper, despite the holiday by January 10, 2022, within 2 days of its delivery. I keep my fingers crossed for the authors in their meaningful research activities.

Reviewer 4 Report

Coatings-1518666

Experimental Study of Black Cotton Soil Stabilization with Natural Lime and Natural Pozzolans as an Alternative to Cut  and Fill in Pavement Subgrade Construction

Comments:

The paper presents information on the stabilization of natural black cotton soil using natural lime and volcanic ashes. The experimental work was designed to study the effect of adding the lime (0-4%), volcanic ash (0-25%), and lime and volcanic ash (1-4, 15-28, respectively) on different engineering properties, i.e. PL, LL, PI, max. dry density, optimum moisture content, California bearing ratio, and swelling. This topic was addressed using the same additives in the cited papers, 24 and 25. The paper lacks novelty and the discussion of the results is mainly dependent on the similar findings in the literature without experimental prove to validate these findings for the studied case, i..e.

  • In section 4.2., line 159 the effect of lime on the MDD was attributed to the particle aggregation and their effect on the soil grading, without providing information from the grading curve and its characteristics.
  • In line 162-165, the behavior was attributed to lower affinity of volcanic ash for water, no experimental investigations were carried out to check if this assumption is valid for this case or not
  • In line 177-178, the behavior of the stabilized soil in the presence of lime was attributed to the pozolanic reaction, this need also to be experimentally proved.

In addition, In page 5 results presented in section 4.1. no explanation was provided to describe the changes in the plasticity of the studied materials, only a description of the trend.

Several grammatical, formatting and typographical errors were detected that needs be corrected, e.g.

In line 43, "Mixing the substance with inherent binder properties and unstable soils to increase ...."

In line 46, “Addictive’s like ordinary Portland cement…”

In page 48, and 65 “.Al-Mukhtar[18,19].”, and “. Zhang et al[7].”

In line 58 “rice hush”

In page 77 “With a CBR value of 1.4” and in table 1 “CBR(%) =1.7 ” pls correct

Table 2 the presented information is chemical composition not properties

Etc

there is a need to insert how the grading curve for BCS was obtained and discuss its characteristics

in line 141, the BCS has acceptable PI (35), without any addition, there is a need to change the text to reflect this

Round 2

Reviewer 1 Report

This manuscript is completely out of the scope of this journal Nothing to do with Coatings, There is no single word about Coating/Coatings in the work Hence reject, and authors may submit to some other journal.  

With Regards,

Author Response

We thank you for your Advice.

Reviewer 2 Report

I recommend to accept the article for printing in present form.

Author Response

We thank you for taking the time to review our article.

Reviewer 3 Report

Title:  Experimental Study of Black Cotton Soil Stabilization with Natural Lime and Pozzolans in Pavement Subgrade Construction.

Journal: coatings

Based on a detailed study of the revised version of the scientific article and the document "Response to Reviewer 3 Comments", I am pleased to say that the assessed contribution to the journal coatings materially, thematically and in terms of content meets my expectations of a quality scientific article. Based on a detailed inspection of the assessed 2nd version, I would like to recommend eliminating the following minor shortcomings, which I did not notice or. forgot to include in the 1st report.

LNSA (Line number of the scientific article) 107-109...Both VA and BCS were oven-dried for 24hrs at 105 ° c temperature. The samples were then weighed and thoroughly mixed with different lime, VA and BCS combination dosage and ready for testing. Design scheme ... The text "Design scheme" is probably extra or the sentence is incomplete.

LNSA 145...Figure 4. CBR swell tripod for measuring swell ... typo measring.

LNSA 150-152…LL of BSC mixed with lime also shows a steady LL decrease; 4% lime stabilized BCS reduces LL from 70.7% to 64.3%. The abbreviation BSC is used only in this sentence, in other cases it is the abbreviation BCS.

LNSA 168-172...Figure 5. Effects of Lime & VA and (b) Lime-VA combined stabilier on the plasticity of BCS ... Mix desgination (%) ... probably Mix designation (%)

LNSA 182...In Figure 6Error! Reference source not found.Error! Reference source not found. (B) ...

LNSA 188-192...Figure 6. Effects of (a) Lime and (b) VA on OMC and MDD of black cotton soil...In Figure 5 there is a 4 times typo "Maximum dry desnsity". The figure uses the description of the y = axis in the form "Maximum dry density (Kg /m3)",  while in Table 1 MDD (kg /m3).  I recommend using the same unit format for kilograms.

LNSA 222...Title Table 5. Subgrade soil classification in Kenya [5] move to the next page of the article and list CBR units in the table.

LNSA 252...[5] Ministry of Public Works Kenya, “Part III - Materials & Pavement Design.pdf,” Ministry of Public Works Kenya.1999.. correct the typo of Kenya.

In conclusion, I would like to thank authors for having incorporated virtually all of my requirements, both optional and mandatory, and I sincerely keep their fingers crossed for them in their scientific activities in the field of scientific assistance to Africa.

Reviewer 4 Report

Coatings-1518666 r2

Experimental Study of Black Cotton Soil Stabilization with Natural Lime and Natural Pozzolans as an Alternative to Cut  and Fill in Pavement Subgrade Construction

The manuscript is considerably increase and is recommended for publication after minor revision to consider the following:

In page 3 figure 3 was mistakenly inserted pls remove it

In page 3 it is preferable to insert the equation and then comment of the values obtained from the calculations

In line 182 remove “Error! Reference source not found.Error! Reference source not found.
